# Improved Catalytic Activity of Spherical Nucleic Acid Enzymes by Hybridization Chain Reaction and Its Application for Sensitive Analysis of Aflatoxin B1

**DOI:** 10.3390/s24072325

**Published:** 2024-04-05

**Authors:** Wenjun Wang, Xuesong Li, Kun Zeng, Yanyan Lu, Boyuan Jia, Jianxia Lv, Chenghao Wu, Xinyu Wang, Xinshuo Zhang, Zhen Zhang

**Affiliations:** 1School of the Environment and Safety Engineering, Jiangsu University, Zhenjiang 212013, Chinakjj80116@ujs.edu.cn (K.Z.); yanyanlu@ujs.edu.cn (Y.L.); 2212209011@stmail.ujs.edu.cn (B.J.); 2222309056@stmail.ujs.edu.cn (X.W.); 3220903047@stmail.ujs.edu.cn (X.Z.); 2National Narcotics Laboratory Beijing Regional Center, Beijing 100164, China; ljx19801128@sina.com (J.L.); vincentwch@163.com (C.W.)

**Keywords:** bioanalysis, signal amplification, rapid detection, aflatoxin B1, hybridization chain reaction

## Abstract

Conventional spherical nucleic acid enzymes (SNAzymes), made with gold nanoparticle (AuNPs) cores and DNA shells, are widely applied in bioanalysis owing to their excellent physicochemical properties. Albeit important, the crowded catalytic units (such as G-quadruplex, G4) on the limited AuNPs surface inevitably influence their catalytic activities. Herin, a hybridization chain reaction (HCR) is employed as a means to expand the quantity and spaces of G4 enzymes for their catalytic ability enhancement. Through systematic investigations, we found that when an incomplete G4 sequence was linked at the sticky ends of the hairpins with split modes (3:1 and 2:2), this would significantly decrease the HCR hybridization capability due to increased steric hindrance. In contrast, the HCR hybridization capability was remarkably enhanced after the complete G4 sequence was directly modified at the non-sticky end of the hairpins, ascribed to the steric hindrance avoided. Accordingly, the improved SNAzymes using HCR were applied for the determination of AFB1 in food samples as a proof-of-concept, which exhibited outstanding performance (detection limit, 0.08 ng/mL). Importantly, our strategy provided a new insight for the catalytic activity improvement in SNAzymes using G4 as a signaling molecule.

## 1. Introduction

Spherical nucleic acids enzymes (SNAzymes) are usually fabricated using functional DNA strands decorated on gold nanoparticles (AuNPs), which are extensively applied in biosensing depending on their ease of synthesis, programmable functionalization, and high stability [1,2,3,4,5]. Compared to other DNA sequences, G-quadruplex (G4) exhibits higher density of the nucleic acid layer after it is modified on the surface of AuNPs through the Au–S covalent band [6], showing good peroxidase-mimicking activity with the presence of the cofactor hemin [7,8,9,10]. However, G4 was formed via the Hoogsteen base pair instead of the traditional Watson–Crick base pair, which occupied more spaces on the limited surface of gold core [11,12,13], resulting in poor catalytic capacity due to the decreased local catalytic substrate concentrations [14,15,16]. Meanwhile, considering that its catalytic activity originated from the G4 DNAzyme layer rather than AuNPs core [17,18], an improvement regarding overcrowding G4 units on the limited AuNPs surface is urgently needed.

In addition, the G4 DNAzyme sequence was popularly applied in a bioassay as a signal molecule, functionalized with a classical 3:1 split mode at its 5′ to 3′ on hairpins’ sticky ends in the process of a hybridization chain reaction (HCR) and further achieved better sensitivity in detecting targets by expanding and evenly distributing catalytic units [19,20,21]. The sequence of sticky ends in hairpin structures plays a crucial role in the performance of HCR. In particular, the effective initiation of two hairpins is hindered by the presence of steric hindrance from the G4 sequence within the toehold domain, leading to low hybridization products and restricted catalytic capacity [22]. To circumvent this problem, the functional DNA sequences for amplification should be preferentially linked to the non-sticky end of the hairpins, which will avoid interfering with hybridization. For example, Ning et al. modified the poly-A sequence at the non-sticky end of the hairpin probe to connect AuNPs [23], effectively triggering the HCR reaction along with satisfactory products. Similarly, Sun et al. possessed an overhang complementary sequence of the aptamer at the 5′ end of the hairpins (non-sticky end) [24], enriching the quantity of the aptamer on the HCR nanowires without affecting the hybridization performance due to the absence of additional strands at the sticky end of the hairpins.

Inspired by the studies in the literature mentioned, in the present study, the catalytic performance impact of SNAzymes was systematically investigated by modulating steric hindrance from the G4 sequences split proportion of the hairpins stem region with HCR-triggered signal amplification. To validate the feasibility of our proposed strategy, aflatoxin B1 (AFB1), as one of the most toxic mycotoxins, was chosen as a model analyte, and we then employed a DNA machine to produce them as catalytic nano-labels in situ for AFB1 detection captured by aptamer in food samples (Figure 1). Unlike the traditionally used method of directly modifying the G4 sequence onto the surface of AuNPs, HCR not only greatly expanded the number of G4 units but also distributed the G4 DNAzymes more evenly to reduce steric hindrance, ultimately leading to better catalytic activity.

## 2. Materials and Methods

### 2.1. Materials and Reagents

All DNA sequences (shown in Appendix A) were synthesized by Sangon Biotech. Co., Ltd. (Shanghai, China) and purified using high-performance liquid chromatography (HPLC). Tris (2-carboxyethyl), Tris(2-carboxyethyl) phosphine hydrochloride (TCEP), 6× Glycerol Gel Loading Buffer, and 10× TM buffer (80 mM MgSO_4_; pH 7.4) were also obtained from Sangon Biotech. Co., Ltd. (Shanghai, China). Chloroauric acid (HAuCl_4_), NaCl, MgCl_2_, Trisodium citrate, and H_2_O_2_ were purchased from Sinopharm Chemical Reagent Co., Ltd. (Shanghai, China). Hemin, 2-methylimidazole, 3,3′,5,5′-tetramethylbenzidine (TMB), Tween 20, and Tris-HCl buffer (pH 7.4) were provided by Aladdin Biochemical Technology Co., Ltd. (Shanghai, China). The AFB1 ELISA kit was bought from BEOSEN Food Safety Technology Co., Ltd. (Wuxi, China). Aflatoxin B1 (AFB1), deoxynivalenol (DON), zearalenone (ZEN), fumonisin B1 (FB1), aflatoxin B2 (AFB2), and ochratoxin (OTA) were obtained from Pribolab Co., Ltd. (Qingdao, China). Ultrapure water (18.2 MΩ·cm) was prepared through the Millipore water purification system.

### 2.2. Preparation of AuNPs-HCR

The AuNPs of 13 nm were synthesized by typical reduction using the sodium citrate method [25]. Briefly, 50 mL of HAuCl_4_ (1 mM) was heated to boiling with vigorous stirring, and then 5 mL of sodium citrate of 38.8 mM was added quickly with continuous heating for 20 min to achieve AuNPs with a diameter of 13 nm. After that, 4 μL of 100 μM of thiol-modified hairpin (HP) (shown in Appendix A) was incubated with 2 μL of 10 μM of TCEP for 30 min. Subsequently, 500 μL of AuNPs was pipetted into the above mixture and kept at −20 °C in a freezer for 2 h [26]. The resulting solution was annealed at 95 °C for 5 min and kept at room temperature for 30 min to form AuNPs-HP structures. Finally, 20 μL of 5 μM of H1 and H2 strands was added for 2 h to obtain AuNPs-HCR complexes.

### 2.3. Atomic Force Microscopy (AFM) Imaging of the AuNPs-HCR

To begin with, 20 μL of TM buffer (1×) was deposited onto freshly cleaved mica to stand for 5 min, rinsed with ultrapure water at least 5 times, and air-dried with compress nitrogen. Subsequently, 10 μL of samples was separately dropped onto mica surface and incubated for 5 min to complete adsorption, washed with ultrapure water three times, and dried again. The samples were scanned on an AFM (Multimode 8, Bruker, Brerica, MA, USA) in air under intelligent imaging mode, and data were processed using NanoScope Analysis software.

### 2.4. Agarose Gel Electrophoresis Analysis of AuNPs-HCR

Generally, the electrophoresis experiment of AuNPs-HCR was conducted after centrifugation and resuspension twice to obtain the concentration. Then, each sample was mixed with loading buffer at a 5:1 ratio and, in total, 10 μL of this mixture was loaded into individual wells. The electrophoresis was implemented in 1× TBE buffer at 100 V for 40 min, and the images were captured by a fluorescence image system (Fusion-solo 3S, Paris, France).

### 2.5. Polyacrylamide Gel Electrophoresis (PAGE) Analysis

DNA enzymatic cleavage reaction dependent on Mg^2+^ ions was confirmed on a 12% (*w*/*w*) polyacrylamide gel. First, all sequences were annealed by heating at 95 °C for 10 min and kept at room temperature for 30 min. The mixture containing 1 μM of primer strand, 1 μM of I_A_, 1 μM of I_B_, and 1 μM of H3 was then incubated at 37 °C in a cleavage buffer solution (20 mM of MgCl_2_, 250 mM of NaCl, and 0.01% Tween 20) for 2 h. Electrophoresis was performed under a constant voltage of 100 V for 50 min. Finally, the gel was stained with Gel Red for 10 min and analyzed using a gel imaging analysis system (Fusion-solo 3S, Paris, France).

### 2.6. Colorimetric Detection Strategy of AFB1

Typically, Apt-_AFB1_ (10 μL, 1 μM) was mixed with S1 (10 μL, 1 μM) for 30 min. Varying concentrations of target AFB1 were then added into the solution and incubated at 37 °C for an additional 30 min. Afterward, I_A_ (10 μL, 1 μM) and I_B_ (10 μL, 1 μM) were then mixed with the above solution containing the released S1 and maintained at 37 °C. After 40 min, AuNPs-HP was then injected into the mixture at 37 °C for 2 h to trigger an off-the-cleavage reaction. Then, H1 and H2 (20 μL, 2.5 μM) were added and allowed to react at 37 °C for 1 h, resulting in triggering HCR to form AuNPs-HCR SNAzymes. Afterward, the solution was centrifuged at 11,000 rpm for 10 min to remove unbound H1 and H2 and resuspended in 10 mM Tris-HCl buffer. Finally, 50 μL of the above products was taken and added to TMB/H_2_O_2_ solutions. After 10 min of incubation, the absorption spectra were recorded by a microplate reader.

## 3. Results and Discussion

### 3.1. Principle of AuNPs-HCR for AFB1 Detection

A detailed depiction of the SNAzymes-based colorimetric biosensing strategy for AFB1 detection is displayed in Figure 1, comprising three integral parts: (A) the process of recognizing the AFB1 target, (B) the cleavage reaction of Mg^2+^ DNAzyme leading to the formation of AuNPs-I, (C) diagram depicting the catalytic effects of AuNPs-HCR with various hairpin split modes.

The target recognition segment involved two strands (Figure 1A): aptamer strand and complementary sequence S1. When the target AFB1 was present, the aptamer captured AFB1 to release the S1 strand, forming the Mg^2+^-dependent split DNAzyme after the addition of two template probes (I_A_ and I_B_). Figure 1B showed the principle of the DNAzyme cleavage reaction for the growth of AuNPs-I. First, the HP hairpin strands were attached to the surface of gold nanoparticles through a thiol linkage, consisting of the HCR-triggering strand and the magnesium ion cleavage site. In the presence of Mg^2+^, the activity of the Mg^2+^-dependent DNAzyme was activated, resulting in the initiator strand of the hairpins being exposed after the cleavage process. Different split modes of hairpins AuNPs-I were then employed to initiate HCR, thereby forming AuNPs-HCR SNAzymes with varying catalytic capabilities (Figure 1C).

### 3.2. Characterization of AuNPs-HCR

The stepwise self-assembly process of AuNPs-HCR was characterized using TEM, AFM, DLS, UV-vis spectra, Zeta potential, and agarose gel electrophoresis. As shown in Figure 1A, the size of the bare AuNPs was estimated to be around 13 nm from TEM and AFM results, which was consistent with the previous literature [27]. After modification, the gold nanoparticles coated with modified HP showed a dispersed state in TEM, which was attributed to the presence of a protective layer that prevented aggregation and resulted in better dispersion (Figure 1B). In addition, as shown in Figure 1C, AuNPs-HCR exhibited even better dispersion compared to AuNPs-HP, which was in agreement with the DLS and zeta potential results (Figure 1F,G). Based on the peak height from the AFM data in Figure 1A–C, the presence of 2 nm high nanowires around the nanocore was observed, indicating the generation of HCR products on the surface of gold nanoparticles compared to bare AuNPs and AuNPs-HCR (Figure 1D). In addition, UV-vis spectra were also used to monitor the self-growth of AuNPs-HCR. The absorption peak of gold nanoparticles near 520 nm was red-shifted with surface DNA modification (Figure 1E) [28,29]. The characteristic peak of AuNPs-HCR was 525 nm. In contrast, the absorption peaks of bare AuNPs and AuNPs-HP were 522 nm and 523 nm, respectively. At the same time, DNA characteristic peaks near 260 nm were observed, which also proved that HCR products were attached to the surface of AuNPs [30,31]. Moreover, agarose gel electrophoresis also demonstrated the self-growing process of AuNPs-HCR. As the DNA on the surface of AuNPs increases, its electrophoretic movement rate gradually decreases, indicating its successful preparation (Figure 1H). However, bare AuNPs lack the protection of DNA and are prone to self-aggregation in the electrophoresis solution [32]. Further, the color development experiment of the TMB/H_2_O_2_ system was also carried out (Figure 1I), where AuNPs-HCR showed higher absorbance than the G4/hemin alone at the same concentration, which also illustrated its successful synthesis. The above results prove that the AuNPs-HCR was successfully prepared.

### 3.3. Assessment of Catalytic Performance for Different Split Modes

First, the enzymatic activity of AuNPs-HCR was evaluated, wherein hairpins with different splitting modes produce different amounts of G4 enzymes, which could determine the catalytic performance (Figure 2A). Figure 2B–D show the typical UV-vis spectra and corresponding oxidized TMB images after mixing with 3:1, 2:2, and 4:0 split mode hairpins. Interestingly, when different split modes hairpins were triggered by AuNPs-I, the different absorbance of the oxidized TMB was observed. The traditional 3:1 split mode of AuNPs-HCR exhibits lower catalytic activity compared with the 2:2 and 4:0 split mode, illustrating that the distribution of the G4 sequence on the hairpin has an important influence. And the 4:0 split mode of AuNPs-HCR displayed a faster reaction rate within the same period of time, which was ascribed to the excellent activity of AuNPs-HCR triggered by 4:0 hairpins.

To further access different split modes of hairpins on the catalytic activity of AuNPs-HCR, steady-state kinetic experiments were employed (Appendix A) [33,34]. After being fitted by Lineweaver-Burk plots, two key parameters in enzyme catalysis were calculated, termed the Michaelis-Menten constant (K_m_) and the maximal reaction velocity (V_max_). As displayed in Figure 2F, the K_m_ values of 2:2, 3:1, and 4:0 split modes were 0.67, 0.91, and 0.32, respectively, indicating that DNA-HCR with a 4:0 split mode had a better affinity to the TMB substrate. In addition, the V_max_ of the corresponding AuNPs-HCR with a 4:0 split mode was higher than others, which further illustrated the faster reaction rate of the catalytic process. In order to more intuitively understand the significance of adjusting different split ratios of hairpins, we compared their catalytic performance with traditional SNAzymes and different concentrations of the G4 DNAzyme (Figure 2G). As expected, the absorbance of AuNPs-HCR with the 4:0 split mode was approximately equal to 230 units of G4 DNAzyme when substituted into the linear equation (Appendix A). Notably, the absorbance of the optimal ratio of AuNPs-HCR is twice that of traditional SNAzymes at the same concentration (1 nM), indicating that the AuNPs-HCR with a 4:0 split mode has the best catalytic activity. Taken together, different split modes of hairpins have a vital impact on the number of G4 DNAzymes, which ultimately influence the catalytic activity of SNAzymes.

### 3.4. Investigation of Catalytic Activity Enhancement

The core component of HCR hairpins was composed of three parts: toehold, stem, and loop [35]. In our experiment, the components of the stem and loop were unchanged, while different G4 split modes led to changes in the length of toehold, which in turn altered the efficiency of the HCR due to additional steric hindrance from the G4 sequence at the sticky end of the hairpins (Figure 3A). A relatively low concentration of initiating strands triggered the reaction until the hairpins were exhausted, which demonstrated that HCR has a high reaction efficiency [36,37]. Thus, we can optimize the concentration of initiator strand to investigate the hybridization rate of hairpins with different split modes. As shown in Figure 3B, Lanes 1–4 represent a hairpin generation split mode of 3:1. As the concentration of the priming strand increases, the band intensity slightly increases, indicating minimal HCR product generation. The 2:2 split mode showed a significant increase in band intensity, but the bands were relatively concentrated and not long, indicating that the HCR reaction efficiency was generally average (Lane 5–8). In the 4:0 split mode, when the initiator strand was present (Lane 9–12), bright and long bands indicated that significant HCR products were present with high efficiency. Meanwhile, in order to visually observe the efficiency of the HCR reaction, we conducted a grayscale analysis using ImageJ software (Version 1.54) and obtained a yield by comparing the grayscale value of the HCR products to that of the total reactants. As shown in Figure 3C–E, the reaction efficiency was the lowest in the 3:1 split mode, while the 2:2 cleavage mode fluctuates around 50% yield. In contrast, the 4:0 mode maintained an 80% yield at lower concentrations of initiator strand, suggesting that the cleavage ratio of HCR had a significant impact on the efficiency of the HCR reaction.

To better understand the distribution of the toehold, stem, and loop structures of different split modes, we simulated their secondary structures using NUPACK (Appendix A). As shown in Figure 4A, in both the 3:1 and 2:2 modes, the sticky ends that were triggered by hairpins have sequences of G4 linkers, but the sticky ends of the 4:0 mode were not. Therefore, we note that it is this additional sequence that increases the steric hindrance during hybridization, resulting in low hybridization efficiency. However, as shown in Figure 3B, the hybridization efficiency of 2:2 is higher than that of 3:1, but the length of its sticky end was longer than that of 3:1. We suspected that it may form into a secondary structure, which will effectively reduce steric hindrance. To prove our conjecture, we simulated the secondary structure of its 2:2 sequence and found that it can form a G-triplex structure through Hoogsteen-like hydrogen bonds (Figure 4B) [38].

Finally, AuNPs-HCR assemble on the surface of AuNPs with different split modes to form SNAzymes catalytic structures. Appendix A shows the TEM of SNAzymes in three split modes, and it can be seen that the 4:0 mode has a larger gap than the 2:2 mode and the 3:1 mode, proving that it had a larger catalytic layer. In addition, through AFM images (Appendix A), it can also be seen that the 4:0 mode has a higher height than the other two modes, agreeing with the TEM results. Thus, the hairpins under the 4:0 split mode could effectively trigger the HCR reaction due to its low steric hindrance, allowing the SNAzymes to have a larger catalytic layer with a higher catalytic capacity.

### 3.5. Analytical Performance of AuNPs-HCR for AFB1 Detection

To demonstrate the application of this robust and stable absorbance indicator, a DNA walker was constructed utilizing AuNPs-HCR for label-free target detection. As depicted in Figure 1A, Apt-_AFB1_ and S1 were hybridized to form hybrid duplexes. Upon the addition of AFB1, Apt-_AFB1_ specifically interacted with AFB1, thereafter releasing S1. Aptamer binding ability was validated via agarose electrophoresis and a CD experiment. The results of from electrophoresis (Appendix A) showed that as the concentration of AFB1 gradually increased, the bands of complementary strands to the aptamer became increasingly visible (Lane 5–7), indicating that AFB1 had successfully bound to the aptamer. Additionally, as seen from Appendix A, AFB1 alone did not have any obvious characteristic peaks, while the binding of AFB1 to the aptamer had a significantly larger absolute value at the 280 nm peak and 240 nm valley than the aptamer alone after the addition of AFB1, which suggested the formation of AFB1 and aptamer conjugate.

We next verify the formation and function of Mg^2+^ DNA enzyme via polyacrylamide gel electrophoresis (PAGE). From Figure 5A, the presence of a new band was observed when the released S1 was present, indicating the formation of a circular unit (Lane 4). When all components participated in the reaction, a new complete band appeared, demonstrating the formation of the composite structure (Lane 10). By comparing Lane 10 and Lane 11, only the DNAzyme structure formed by the probe at the rA site produced a new band, which coincided with the expected position of the product chain, confirming successful digestion. The above gel electrophoresis results indicate that the DNA walker constructed using AuNPs-HCR had successfully enabled precise identification and multi-stage amplification.

After optimization of the experimental conditions (Appendix A), our proposed strategy was employed for the detection of the target molecule AFB1 (one of the most toxic mycotoxins) [39,40]. Under the optimal conditions, the limit of detection (LOD) was reached as low as 0.08 ng/mL (S/N = 3) (Figure 5C,D), which was lower than other methods (Appendix A). Furthermore, AuNPs-HCR as the label with 4:0 split mode exhibited the highest absorbance in comparison to 3:1 and 2:2 split modes, showcasing the remarkable sensing performance of the AuNPs-HCR-labeled colorimetric sensor (Figure 5E). To investigate the specificity of the proposed methods for the detection of AFB1, five other toxins, including DON, ZEN, FB1, AFB2, and OTA, were chosen as interferents. As shown in Figure 5F, the AFB1 and mixture exhibited obvious high absorbance compared with the other five molecules, revealing good selectivity for the detection of AFB1 in complex samples. In addition, the relative standard deviations (RSDs) of intra-assay and inter-assay about our proposed method ranged from 3.98 to 8.10% and from 3.31 to 8.47%, respectively, indicating good repeatability (Appendix A).

### 3.6. Application of AuNPs-HCR in Real Samples

To further validate the application and feasibility of the proposed method for AFB1 detection, five different food samples (corn, red bean, rice, wheat, peanut) were chosen for analysis (sample preparation shown in Appendix A) [41,42]. As displayed in Appendix A, the recoveries of AFB1 in food samples varied from 96.5% to 107.4%, with coefficient variation from 2.67% to 5.37%, indicating that the proposed method could be employed in real food samples. Additionally, the accuracy of the developed method was also verified by commercial ELISA kits. The test results are shown in Appendix A with acceptable t_exp_ using Student’s classic *t* test (lower than 2.78), demonstrating good accuracy compared to commercial methods [43]. The results presented above indicate that the suggested approach demonstrated a satisfactory performance in identifying AFB1 in actual food samples.

## 4. Conclusions

In summary, HCR was employed as an effective tool to expand the number of G4 catalytic units on the limited AuNPs surface, but also to reasonably separate the positions of G4 strands to avoid steric hindrance, thereby improving the catalytic activity of SNAzymes. As a label for HCR signal amplification, the position of G4 sequence decorated on the hairpins will obviously influence the HCR hybridization activity. When the G4 sequence is in the 4:0 mode, the hybridization efficiency is the highest due to the sequence modification at the non-sticky ends of DNA hairpins, which effectively reduces steric hindrance. In addition, its application for AFB1 analysis demonstrated its good performance (excellent sensitivity, accuracy, and stability), suggesting great potential for rapid detection against varied targets. Considering that the proof of the G4 structure in different sequences is not yet complete, only structural simulations have been conducted without experimental verification. Therefore, we need to further address these deficiencies in our future work.

## Data Availability

Data will be made available on request.

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
