# Peer review of "Improved Catalytic Activity of Spherical Nucleic Acid Enzymes by Hybridization Chain Reaction and Its Application for Sensitive Analysis of Aflatoxin B1"

_sensors, 2024, doi:10.3390/s24072325_

Round 1
Reviewer 1 Report
Comments and Suggestions for Authors
In this study, hybridization chain reaction was successfully used to improve the catalytic activity of spherical nucleic acids enzyme (SNAzyme). The effectiveness of SNAzyme has been confirmed in a colorimetric method for the highly sensitive detection of aflatoxin B1. The work describes many experimental studies and impressive results. However, it is worth noting the negligence in the description of the data obtained and incorrect formulations, which make it difficult to understand the essence of the work. Below are a number of comments to the authors:
1) Please adjust the position of bibliography references throughout the text.
2) There are incorrectly formulated phrases in the manuscript. For example: “In which, the sequence of hairpins sticky ends have vital impacts on HCR performance, and its two hairpins cannot be effectively initiated due to additional steric hindrance of G4 sequence from toehold domain, leading to low hybridization products and restricted catalytic capacity”.
3) Line 72. The full name of TCEP is Tris(2-carboxyethyl)phosphine hydrochloride. Please correct it in the manuscript.
4) Line 87-88. What is thiol- modified HP? A reference to supplementary materials should be provided here.
5) In the caption to Figure 2 there is no transcript of the E-G data.
6) Line 299, the limit of detection instead of “the limited of detection”.
7) The captions for Figure 5 do not reflect the data presented in the figure. From Figures 5 C and D, it is difficult to estimate the detection limit calculated by the authors as 0.08 ng/mL. It should be clarified what concentrations of aflatoxin B1 correspond to the absorption spectra shown in Figure 5 C.
8) What explains the high specificity of the developed method for aflatoxin B1 compared to its close analogue aflatoxin B2? (line 304-305, Figure 5F).
9) After determining the analytical characteristics of the developed method, the achieved detection limit and analysis time should be compared with other colorimetric methods for the determination of aflatoxin B1 previously described in the literature. Please add this information to your manuscript.
Author Response
Reviewer 1
Comments:
In this study, hybridization chain reaction was successfully used to improve the catalytic activity of spherical nucleic acids enzyme (SNAzyme). The effectiveness of SNAzyme has been confirmed in a colorimetric method for the highly sensitive detection of aflatoxin B1. The work describes many experimental studies and impressive results. However, it is worth noting the negligence in the description of the data obtained and incorrect formulations, which make it difficult to understand the essence of the work. Below are a number of comments to the authors:
- Please adjust the position of bibliography references throughout the text.
Answer: Done.
- There are incorrectly formulated phrases in the manuscript. For example: “In which, the sequence of hairpins sticky ends have vital impacts on HCR performance, and its two hairpins cannot be effectively initiated due to additional steric hindrance of G4 sequence from toehold domain, leading to low hybridization products and restricted catalytic capacity”.
Answer: Thank you very much for your valuable comment. This sentence has been rewritten according to your good suggestions (Line 47-51).
- Line 72. The full name of TCEP is Tris(2-carboxyethyl) phosphine hydrochloride. Please correct it in the manuscript.
Answer: Done.
- Line 87-88. What is thiol- modified HP? A reference to supplementary materials should be provided here.
Answer: Thiol- modified HP is the modification of SH at the 5’ of Hairpins, which aims to link with AuNPs via Au-S bound, and the reference you mentioned also was supplemented.
- In the caption to Figure 2 there is no transcript of the E-G data.
Answer: Sorry for this mistake due to our carelessness, we have added the captions of the Figure 2 E-G.
- Line 299, the limit of detection instead of “the limited of detection”.
Answer: Done.
- The captions for Figure 5 do not reflect the data presented in the figure. From Figures 5 C and D, it is difficult to estimate the detection limit calculated by the authors as 0.08 ng/mL. It should be clarified what concentrations of aflatoxin B1 correspond to the absorption spectra shown in Figure 5 C.
Answer: Very sorry for this confusion because we uploaded the wrong captions, which now was corrected and displayed at Figure 5C (the revised version).
- What explains the high specificity of the developed method for aflatoxin B1 compared to its close analogue aflatoxin B2? (line 304-305, Figure 5F).
Answer: Thanks a lot for your good comment. Aptamers are single stranded DNA or RNA sequences with unique secondary and tertiary structures for capturing targets, which have advantages in specific recognition along with high affinity. Although AFB1 and AFB2 are similar in structure, their aptamer sequences are different, resulting in different affinities against corresponding targets, even if they are analogs. (Food Chem., 2024, 436, 137661; Food Control., 2015, 45, 545). Therefore, the AFB1 aptamer used in this work has good specificity (Figure 5F).
- After determining the analytical characteristics of the developed method, the achieved detection limit and analysis time should be compared with other colorimetric methods for the determination of aflatoxin B1 previously described in the literature. Please add this information to your manuscript.
Answer: Thank you very much for your good comment, and the colorimetric methods you mentioned has been supplemented in the Table S5 for comparison with our proposed method.

Reviewer 2 Report
Comments and Suggestions for Authors
G4 DNAzyme sequence exhibits superior peroxidase-like activity, and has a wide range of potential applications in bioanalysis and biomedicine. The manuscript presents the effects of different split modes on the hybridization chain reaction were, and SNAzymes was applied in the detection of AFB1. Overall, this manuscript is well organized and written, there are a little left for improvement before considering publication. The following concerns should be noted.
This manuscript introduces the application of HCR in the detection of AFB1, however, in the introduction section, there is no introduction of any AFB1 and aptamer related, please supplement this.
The illustration of Figure 5 is completely the same as Figure 4, please check it.
In the Conclusions and future perspective, the authors’ view on the future development is lacking. With the abundant knowledge of authors in this field, could please authors point the current shortage. Readers and researchers are expected to see more clear direction in the further development and applications of this technology by reading this manuscript.
Comments on the Quality of English LanguageLine 72, add space before “(”.
Line 105,113, add space before “V”.
Line 108, “2+” should be superscript.
Add space before each “℃”.
Line 167, two references should be placed in one parenthesis.
Line 178, add space after “)”.
Line 183, “2” should be subscript.
Line 211, add space before “nM”.
Line 266-268, add space before “μM”.
Line 300, add space before and after “=”.
Author Response
Comments:
G4 DNAzyme sequence exhibits superior peroxidase-like activity, and has a wide range of potential applications in bioanalysis and biomedicine. The manuscript presents the effects of different split modes on the hybridization chain reaction were, and SNAzymes was applied in the detection of AFB1. Overall, this manuscript is well organized and written, there are a little left for improvement before considering publication. The following concerns should be noted.
- This manuscript introduces the application of HCR in the detection of AFB1, however, in the introduction section, there is no introduction of any AFB1 and aptamer related, please supplement this.
Answer: According to your valuable suggestions, some relevant descriptions have been supplemented in the introduction (Line 63-66).
- The illustration of Figure 5 is completely the same as Figure 4, please check it.
Answer: We are so sorry for this confusion due to our carelessness, and now it has been added.
- In the Conclusions and future perspective, the authors’ view on the future development is lacking. With the abundant knowledge of authors in this field, could please authors point the current shortage. Readers and researchers are expected to see more clear direction in the further development and applications of this technology by reading this manuscript.
Answer: Thanks for reminding us to offer the current shortages of our work, and now this part you mentioned was provided in the conclusion (Line 345-347).
- Line 72, add space before “(”.
Answer: Done.
- Line 105,113, add space before “V”.
Answer: Done.
- Line 108, “2+” should be superscript.
Answer: Done.
- Add space before each “℃”.
Answer: Done.
- Line 167, two references should be placed in one parenthesis.
Answer: Done.
- Line 178, add space after “)”.
Answer: Done.
- Line 183, “2” should be subscript.
Answer: Done.
- Line 211, add space before “nM”.
Answer: Done.
- Line 266-268, add space before “μM”.
Answer: Done.
- Line 300, add space before and after “=”.
Answer: Done.

Reviewer 3 Report
Comments and Suggestions for Authors
In this article, Wenjun Wang et al. used the improvement of the catalytic activity of spherical nucleic acid enzymes in conjunction with the hybridisation chain reaction for the specific detection of aflatoxin. They did a great job and optimised all parameters in the different steps of the design. Finally, they managed to achieve a good detection limit for aflatoxin and a high selectivity towards other toxins. I think this work is of interest to the journal and recommend its publication with only two comments:
- Figure 1 is a little difficult to understand. I would suggest changing the colours or trying to redesign it so that it is more friendly and can be understood by non-experts in the field.
- In Figure 2, the caption of Figures E to G is missing.
Author Response
In this article, Wenjun Wang et al. used the improvement of the catalytic activity of spherical nucleic acid enzymes in conjunction with the hybridisation chain reaction for the specific detection of aflatoxin. They did a great job and optimised all parameters in the different steps of the design. Finally, they managed to achieve a good detection limit for aflatoxin and a high selectivity towards other toxins. I think this work is of interest to the journal and recommend its publication with only two comments:
- Figure 1 is a little difficult to understand. I would suggest changing the colours or trying to redesign it so that it is more friendly and can be understood by non-experts in the field.
Answer: Thank you very much for this good suggestion. We have changed some colors of Figure 1 to make it more friendly.
- In Figure 2, the caption of Figures E to G is missing.
Answer: We are so sorry for this confusion due to our carelessness, and now the caption of Figure 2E-G has been added.

Round 2
Reviewer 1 Report
Comments and Suggestions for Authors
The authors took into account all comments and improved the quality of the manuscript.